# An Enhanced Flower Pollination Algorithm with Gaussian Perturbation for Node Location of a WSN

**DOI:** 10.3390/s23146463

**Published:** 2023-07-17

**Authors:** Jun Zheng, Ting Yuan, Wenwu Xie, Zhihe Yang, Dan Yu

**Affiliations:** 1College of Optical, Mechanical and Electrical Engineering, Zhejiang A&F University, Hangzhou 311300, China; zhengjun@zafu.edu.cn; 2School of Information Science and Engineering, Hunan Institute of Science and Technology, Yueyang 414006, China; 812211120176@vip.hnist.edu.cn (T.Y.); kjc@hnist.edu.cn (Z.Y.); 12003024@hnist.edu.cn (D.Y.)

**Keywords:** wireless sensor network, flower pollination algorithm, Gaussian perturbation, node location, optimization

## Abstract

Localization is one of the essential problems in internet of things (IoT) and wireless sensor network (WSN) applications. However, most traditional range-free localization algorithms cannot fulfill the practical demand for high localization accuracy. Therefore, a localization algorithm based on an enhanced flower pollination algorithm (FPA) with Gaussian perturbation (EFPA-G) and the DV-Hop method is proposed.FPA is widely applied, but premature convergence still cannot be avoided. How to balance its global exploration and local exploitation capabilities still remains an outstanding problem. Therefore, the following improvement schemes are introduced. A search strategy based on Gaussian perturbation is proposed to solve the imbalance between the global exploration and local exploitation search capabilities. Meanwhile, to fully exploit the variability of population information, an enhanced strategy is proposed based on optimal individual and Lévy flight. Finally, in the experiments with 26 benchmark functions and WSN simulations, the former verifies that the proposed algorithm outperforms other state-of-the-art algorithms in terms of convergence and search capability. In the simulation experiment, the best value for the normalized mean squared error obtained by the most advanced algorithm, RACS, is 20.2650%, and the best value for the mean distance error is 5.07E+00. However, EFPA-G reached 19.5182% and 4.88E+00, respectively. It is superior to existing algorithms in terms of positioning, accuracy, and robustness.

## 1. Introduction

Over recent decades, as the technologies of the internet of things (IoT) have become more and more sophisticated, the intelligent perception and management of objects can be realized through the connection of things and people [1]. Wireless sensor networks (WSNs) have been playing an increasingly significant role in the IoT with their functions such as real-time sensing, collecting, and processing of information. The inherent characteristics of the node location make it an essential prerequisite for many functions. With the demand for location-based services developing rapidly, the accuracy of node localization affects numerous practical application areas, such as smart homes [2], city surveillance [3], fault diagnosis [4], etc.

The localization methods can be classified into range-based [5] and range-free [6] localization methods based on whether the actual measurement of the distance between nodes is necessary. The former achieves the estimation of the distance between nodes based on the angle of arrival [7], time of arrival [8], and time difference of arrival [9] of the signal or based on the received signal strength. Although this type of method has high positioning accuracy, it requires additional equipment to achieve the corresponding measurement, which is costly and not suitable for monitoring of large areas. The latter method estimates the location of unknown nodes by the connectivity between neighboring nodes. Although the accuracy of this type of algorithm is low, it can basically meet the practical needs. This paper focuses on the DV-Hop method, which is one of the typical range-free based localization methods [10]. To satisfy the increasing demand of positioning accuracy, many scholars have conducted research related to the DV-Hop method. For example, Xue [11] proposed an improved DV-Hop algorithm based on hop refinement and distance correction for the shortcomings of the traditional DV-Hop-based wireless sensor network localization algorithm with large errors. Messous et al. [12] proposed an improved recursive DV-Hop localization algorithm by minimizing the localization error of the estimated distance between the anchor node and the unknown node. Cheikhrouhou et al. [13] proposed an enhanced DV-Hop method by transforming the localized nodes into anchor nodes to improve the localization accuracy. Messous et al. [14] proposed an improved DV-Hop method by receiving the signal strength and polynomial approximation to estimate the distance between the anchor node and the unknown node. Zhao et al. [15] proposed a DV-Hop algorithm based on locally weighted linear regression (LWLR-DV-Hop), using a kernel approach to improve localization accuracy by increasing the weights of neighboring anchor nodes. Liouane et al. [16] proposed an improved method of the DV-Hop algorithm for wireless sensor networks based on the Tikhonov regularization method. Liu et al. [17] proposed an improved DV-Hop algorithm based on neural dynamics (ND-DV-Hop) for improving the accuracy of the DV-Hop algorithm. Although the above algorithms can obtain satisfactory localization accuracy, with the continuous development of metaheuristic optimization algorithms, such as the cuckoo search algorithm (CS) [18], beluga whale optimization (BWO) [19], and golden jackal optimization (GJO) [20], how to better utilize the advantages of both methods is a hot research topic.

The flower pollination algorithm (FPA) [21], as one of the typical metaheuristic optimization algorithms, has attracted much attention by virtue of its effective applicability to real-world problems. Although the FPA can achieve satisfactory performance in solving regular problems, for complex problems it remains limited by its low search capability and convergence speed [22]. Therefore, scholars have conducted much innovative research. Kaya [23] proposed a quick flower pollination algorithm based on parameter adaptive and arithmetic crossover. Cao et al. [24] proposed a robot calibration method using an extended Kalman filter (EKF) and an artificial neural network (ANN) based on the butterfly and flower pollination algorithms (ANN-BFPA) to improve the absolute pose (position and orientation) accuracy of the robot. Ozsoydan et al. [25] proposed a species-based flower pollination algorithm with increased selection pressure and enhanced reinforcement in abiotic local pollination. Mergos [26] applied the FPA for the first time to compute challenging optimal designs of real-world three-dimensional reinforced concrete (RC) building frame structures after a series of appropriate modifications. Dao et al. [27] proposed to hybridize the FPA with a sine–cosine algorithm (called HSFPA) to avoid the drawbacks of the FPA for microgrid operation planning and global optimization problems. Sasikumar et al. [28] proposed a population intelligence-based approach for op-amp optimization sizing. A hybrid version of the flower pollination algorithm (HFPA) is introduced to effectively solve the analog circuit transistor sizing problem and reduce the design search space. Ozsoydan et al. [29] introduced several FPA modifications using chaotic maps and further enhanced the exploited search capabilities by using an enhanced step quantification procedure. All the mentioned studies relevant to the FPA have achieved satisfactory results. However, most of them are parametric algorithms and hybrid algorithms that lack innovative research on the FPA’s search mechanism. Therefore, this paper focuses on relevant studies on global exploration and local exploitation search mechanisms.

With respect to the mentioned problems, an enhanced flower pollination algorithm with Gaussian perturbation (EFPA-G) is proposed in this paper. In the global exploration of the traditional FPA, the optimal search process of the population is achieved only by relying on Lévy flight, which can mean the algorithm becomes trapped in local extremes in the late iterations. Therefore, in the FPA proposed in this paper, an enhanced strategy based on Lévy flight is proposed to improve the search capability of the algorithm. An imbalance in the search capability exists between the global exploration and local exploitation of the traditional FPA, which results in the local exploitation struggling to contribute to a faster convergence speed of the algorithm during the iterations. Therefore, this paper proposes a Gaussian perturbation strategy for improving the search ability of the local exploitation. Extensive experimental results on the benchmark optimization problems show that the proposed EFPA-G algorithm outperforms other state-of-the-art improved metaheuristic optimization algorithms. Satisfactory localization accuracy is also achieved in the WSN simulation experiments.

The major contributions of this article are stated as follows:(1)A localization algorithm based on the enhanced flower pollination algorithm with Gaussian perturbation and the DV-Hop method is proposed. The superiority and robustness of the proposed algorithm are verified by extensive simulation experiments.(2)An enhancement strategy based on Lévy flight is proposed to improve the search capability of the algorithm.(3)A Gaussian perturbation strategy for balancing global exploration and local exploitation search capabilities is proposed.

The remainder of this paper is presented as follows: The FPA and the DV-Hop method are described in Section 2. The proposed EFPA-G algorithm is described in detail in Section 3. The simulation results and comparisons of the approaches are shown in Section 4. Finally, conclusions are presented in Section 5.

## 2. Background

### 2.1. DV-Hop Method

The distance vector hop (DV-Hop) localization algorithm is based on the distance vector exchange protocol to obtain hop counts for node localization. The DV-Hop algorithm is widely used in WSN node localization because of its relatively simple implementation [30], which only requires the use of hop counts and distance vector estimates to calculate the nodes’ positions. No additional hardware or measurement costs are necessary, making it easy to implement and apply in large-scale wireless sensor networks, including dense and sparse networks. Additionally, it can handle situations such as partial node failures compared to other algorithms. The algorithm uses hop counts and distance vector estimates to calculate the nodes’ positions, which can be measured and transmitted by multiple nodes, thereby improving the measurement’s reliability and accuracy. Combining the proposed algorithm with the DV-Hop algorithm increases the algorithm’s robustness and improves its localization accuracy.

#### 2.1.1. Calculating the Minimum Number of Hops

The anchor node broadcasts its own packet to the whole network by flooding, and the packet contains the anchor node’s location information and hop value. The unknown node receives the packet from the anchor node and updates its hop count information, adds 1 to the packet’s hop value and saves it. To ensure that all nodes in the network have access to the minimum hop count $h_{ui}$ information of each anchor node, the node receiving the packet only needs to keep the packet with the smaller hop count.

#### 2.1.2. Estimating the Distance between Anchor Nodes and Unknown Nodes

After each anchor node receives the coordinates of other anchor nodes and the minimum number of hops, it calculates its average hop distance according to Equation (Equation 1).
(1)$$Hopsize_{i}=\frac{\sum_{i\neq j}\sqrt{{\left(x_{i}-x_{j}\right)}^{2}+{\left(y_{i}-y_{j}\right)}^{2}}}{\sum_{i\neq j}h_{ij}}$$
where $\left(x,y\right)$ represents the coordinates of the anchor nodes, $h_{ij}$ represents the number of hops between anchor nodes *i* and *j*, and $Hopsize_{i}$ represents the average hop distance of anchor node *i*.

#### 2.1.3. Estimating Unknown Node Coordinates

The estimated distance $d_{ui}$ is calculated using Equation (Equation 1), and then combined with the least squares method to estimate the position of the unknown node, as shown in Equation (Equation 2).
(2)$$\left\{\begin{array}{c} {\left(x_{1}-x\right)}^{2}+{\left(y_{1}-y\right)}^{2}=d_{1}^{2}\\ {\left(x_{2}-x\right)}^{2}+{\left(y_{2}-y\right)}^{2}=d_{2}^{2}\\ \vdots \\ {\left(x_{n}-x\right)}^{2}+{\left(y_{n}-y\right)}^{2}=d_{n}^{2} \end{array}\right.$$
where $\left(x,y\right)$ denotes the coordinates of the unknown node and *d* denotes the distance between the unknown node and the anchor node. By subtracting the first term from the previous term and splitting it, the form AX=B is obtained, where *A*, *B*, and *X* are shown in Equations (Equation 3)∼(Equation 5), respectively.
(3)$$A=\left[\begin{array}{c} 2\left(x_{1}-x_{n}\right)\begin{array}{cc} \cdots & 2\left(y_{1}-y_{n}\right) \end{array}\\ \begin{array}{ccc} & \vdots & \begin{array}{cccc} & & & \vdots \end{array} \end{array}\\ 2\left(x_{n-1}-x_{n}\right)\begin{array}{cc} \cdots & 2\left(y_{n-1}-y_{n}\right) \end{array} \end{array}\right]$$
(4)$$B=\left[\begin{array}{c} x_{1}^{2}-x_{n}^{2}+y_{1}^{2}-y_{n}^{2}+d_{n}^{2}-d_{1}^{2}\\ \begin{array}{cccc} & & & \vdots \end{array}\\ x_{n-1}^{2}-x_{n}^{2}+y_{n-1}^{2}-y_{n-n}^{2}+d_{n}^{2}-d_{n-1}^{2} \end{array}\right]$$
(5)$$X=\left[\begin{array}{c} x\\ y \end{array}\right]$$

Finally, the estimated coordinates of the unknown nodes are obtained by Equation (Equation 6).
(6)$$\hat{X}={\left(A^{T}A\right)}^{-1}A^{T}B$$

### 2.2. Flower Pollination Algorithm

The flower pollination algorithm (FPA) is a metaheuristic optimization algorithm based on the pollination process in nature. The FPA uses adaptive parameters and adaptive operators, which can dynamically adjust parameters according to the characteristics of the problem and the search process, thereby improving the robustness and adaptability of the algorithm. It is highly robust, and can handle noisy data and complex problems with multiple variables and constraints by using probabilistic methods to explore different regions of the search space. It can quickly converge to a satisfactory solution, reducing calculation time and cost. Because its implementation is relatively simple and can be quickly applied to the solution of various practical problems, it is widely used in various fields such as optimization, classification, and clustering. It mainly consists of global pollination and self-pollination, which can also be called global exploration and local exploitation.

#### 2.2.1. Global Pollination

Global pollination, that is, cross-pollination, is usually based on biology. Pollen is spread over long distances by pollinators such as birds, insects, bees, or bats. Its spread can be modeled using the Lévy distribution, as shown in Equation (Equation 7).
(7)$$x_{i}^{t+1}=x_{i}^{t}+l\overset{´}{e}vy\cdot \left(x_{best}-x_{i}^{t}\right)$$
where $x_{best}$ represents the optimal solution, $x_{i}^{t}$ represents the *i*-th solution of the *t*-th generation, and $l\overset{´}{e}vy$ represents the Lévy flight step length, as shown in Equation (Equation 8).
(8)le´vy∼λΓλsinπλπλ22π·1s1+λs≫s0>0
where $\Gamma \left(\lambda \right)$ represents the standard gamma function, λ=1.5, parameter *s* is calculated using Equation (Equation 9), and μ and *v* are random numbers subject to Gaussian distribution. μ is the distribution with the mean zero variance of $\sigma^{2}$. The parameter σ is calculated using Equation (Equation 10).
(9)s=μv11λλ,μ∼N0,σ2,v∼N0,1
(10)σ=Γ1+λλΓ1+λ1+λ22·sinλπλπ222λ−1λ−12212λ

#### 2.2.2. Local Pollination

Local pollination, that is, self-pollination, is pollination by abiotic and other factors, so the spread range is small, and pollination is often completed around itself, as shown in Equation (Equation 11).
(11)$$x_{i}^{t+1}=x_{i}^{t}+\varepsilon \cdot \left(x_{j}^{t}-x_{k}^{t}\right)$$
where $x_{j}^{t}$ and $x_{k}^{t}$ are pollen from different flowers of similar flowering plants, i.e., two solutions randomly selected from the population, and ε is a random variable that follows a uniform distribution.

#### 2.2.3. Transition Probability

The algorithm randomly switches between the two processes of exploitation and exploration by transition probability *P* to determine the type of flower pollination (i.e., global or local search process) to ensure the quality of the search. When the random probability is less than the transition probability, self-pollination is performed, otherwise the cross-pollination process is performed.

## 3. Proposed EFPA-G Approach

An enhanced flower pollination algorithm with Gaussian perturbation (EFPA-G) is proposed to avoid premature convergence of the FPA algorithm and to improve the performance of the FPA algorithm on complex optimization problems.

### 3.1. Enhanced Strategy

In traditional FPAs, global exploration relies only on Levy flights to carry out global optimization, but the search capability is not sufficient to accomplish high-precision optimization in late iterations. In addition, the Lévy flight only exploits the difference between the optimal solution and the candidate solution, without utilizing the global search information of the population as a whole, resulting in a waste of information. Therefore, an enhanced strategy based on Lévy flight is proposed in this paper to accommodate the requirements of the algorithm for high-precision search optimization in the late iterations, as shown in Equation (Equation 12).
(12)$$x_{i}^{t+1}=x_{i}^{t}+\delta \cdot \left(rand\cdot \left(x_{best}-x_{i}^{t}\right)+l\overset{´}{e}vystep\right)$$
where δ represents the iteration-based weights, as shown in Equation (Equation 13), and $l\overset{´}{e}vystep$ is the leap distance based on the Lévy flight, as shown in Equation (Equation 14).
(13)δ=rand·1−ttIterIter
(14)$$l\overset{´}{e}vystep=\alpha \cdot l\overset{´}{e}vy\cdot \left(x_{k}^{t}-x_{i}^{t}\right)$$
where *t* and Iter denote the current iteration number and the maximum iteration number, respectively. α is a constant, and α=0.05, $x_{k}^{t}$ represents the randomly selected individuals in the *t*-th generation.

### 3.2. Gaussian Perturbation

The enhanced strategy based on Levy flight further improves the search capability of the FPA’s global exploration, but cannot satisfy the needs of large-scale optimization problems, and further increases the imbalance between global exploration and the local development search capability. Therefore, how to enhance the search capability of local exploitation becomes particularly more significant.

In Equation (Equation 11), *x* performs a modification of its own information based only on the information difference with the other two candidate solutions in the population. In addition, the Gaussian distribution alters its curve according to the variation in the mean and variance, which indicates the stability while being random. Therefore, employing a Gaussian distribution in the iteration to achieve perturbation of the population is practicable. In this paper, an optimal search strategy based on Gaussian perturbation is proposed, as shown in Equation (Equation 15).
(15)$$x_{i}^{t+1}=G\left(x_{i}^{t}\right)+\varepsilon \cdot \left(x_{j}^{t}-x_{k}^{t}\right)$$
where $G\left(\cdot \right)$ is defined by Equation (Equation 16).
(16)Gxit=12πζ2e−xit−xbestt2−xit−xbestt22ζ22ζ2
where $x_{best}^{t}$ represents the *t*-th generation optimal solution and ζ is defined by Equation (Equation 17).
(17)ζ=1+logtlogttt·xit−xbest
where $x_{best}$ represents the global optimal solution.

With respect to the designed Gaussian perturbation, it is designed to maintain stability throughout the iterations while further perturbing *x* over a large range, thus permitting the algorithm to leap out of local extremes.

### 3.3. Objective Function

For the proposed EFPA-G algorithm to be more applicable to WSN node localization, the localization problem is transformed into the problem of minimizing the difference between the estimated and actual locations, i.e., the localization error minimization problem, as shown in Equation (Equation 18).
(18)$$\min f\left(x,y\right)=\sum_{i=1}^{N_{a}}\sqrt{a\left[{\left(x-x_{i}\right)}^{2}+{\left(y-y_{i}\right)}^{2}-d_{i}^{2}\right]}$$where $N_{a}$ is the number of anchor nodes, and *a* is the reciprocal of the minimum number of hops between the estimated unknown $\left(x,y\right)$ and the anchor node $\left(x_{i},y_{i}\right)$. $d_{i}^{2}$ denotes the actual distance between the unknown node and the anchor node.

### 3.4. EFPA-G Algorithm

According to the previous theory, this section focuses on detailing the framework of the proposed enhanced flower pollination algorithm with Gaussian perturbation, as shown in Algorithm 1.
**Algorithm 1** Proposed EFPA-G Algorithm**Input:**  The maximum number of iterations Iter, population size *N*, transition probability *P*;**Output:**  Optimal solution;1:Initialize: $x_{i}^{0}=x_{\min}+rand\cdot \left(x_{\max}-x_{\min}\right)$;2:Calculate the fitness of the initial population $x_{i}^{0}$;3:Record the current optimal solution best;4:t=1;5:**while** 
t≤Iter 
**do**6:   **for** i=1 to *N* **do**7:     **if** rand<P **then**8:        $\delta =rand\cdot \left(1-t/Iter\right)$;9:        Update newx using Equation (Equation 12);10:     **else**11:        β=logtlogttt·best−xit;12:        Update newx using Equation (Equation 15);13:     **end if**14:     Calculate the fitness of newx;15:     **if** $fit\left(newx\right)<fit\left(x_{i}^{t-1}\right)$ **then**16:        $x_{i}^{t}=newx$;17:     **end if**18:     **if** $fit\left(newx\right)<fit\left(best\right)$ **then**19:        best=newx;20:     **end if**21:   **end for**22:   t=t+1;23:**end while**24:output result.

### 3.5. Complexity Analysis

This section focuses on the complexity analysis in two parts, the DV-Hop method and the EFPA-G algorithm. In the DV-Hop method, the total number of nodes is *n* and the number of anchor nodes is *m*. The complexity of the minimum hop count estimation between nodes is $O\left(n^{3}\right)$, the complexity of the actual distance calculation between nodes is $O\left(n\times m\right)$, the complexity of the distance between unknown nodes and anchor nodes is N, and the complexity of the likelihood estimation method is $O\left({\left(n-m\right)}^{4}\right)$. In the EFPA-G algorithm, the dimension of the objective function is *D*. The complexity of the proposed algorithm is $O\left(Iter\times N\times D\right)$. By replacing the likelihood estimation method in DV-Hop by the proposed EFPA-G algorithm, the complexity of the final WSN node localization method is $O\left(n^{3}\right)+O\left(n\right)+O\left(n\times m\right)+O\left(Iter\times N\times D\right)$.

## 4. Numerical Results and Discussions

The performance of the proposed algorithm is verified using 26 well-known benchmark functions, and the superiority of the proposed algorithm is confirmed by comparison with other state-of-the-art metaheuristic optimization algorithms. In addition, the performance of the EFPA-G algorithm is tested in a WSN simulation for real optimization problems.

### 4.1. Parameters Settings

In this section, we discuss the performance of the proposed EFPA-G and other state-of-the-art optimization algorithms as follows.

(1)Flower pollination algorithm (FPA): transition probability P=0.5;(2)Flower pollination algorithm based on cloud mutation (CMFPA) [31]: transition probability P=0.5;(3)Modified particle swarm optimization (MPSO) [32]: inertia weight ω=0.9∼0.4, acceleration factors $c_{1}=c_{2}=2$;(4)Marine predators algorithm (MPA) [33]: fish aggregating devices FADs=0.2, constant number p=0.5;(5)Ranking-based adaptive cuckoo search algorithm (RACS) [34]: the maximum size of the archive AN=N, exponent parameter $\alpha_{\min}=0.5,\alpha_{\max}=2.5$, the pre-determined number of cycles limit=N·D, initial crossover rate CRm=0.5;(6)Proposed EFPA-G algorithm: transition probability P=0.5.

For the sake of fairness, the mentioned algorithms are all run 30 times independently with the same population size N=50 and the termination condition is the maximum number of iterations. The maximum number of iterations is set to 1000 for the numerical experiments and to 200 for the WSN simulation experiments.

### 4.2. Benchmark Test

To test the proposed EFPA-G apporach, 26 benchmark functions are available which are divided into unimodal functions ($f_{1}$∼$f_{12}$) and multimodal functions ($f_{13}$∼$f_{26}$). The former reveals the exploitation performance of the algorithm, while the latter challenges the exploration capability of the algorithm. Details of the benchmark functions are shown in Table 1. In addition, to test the optimization performance of the proposed algorithm for benchmark functions of different dimensions, experiments are conducted using D=30 and D=50.

#### 4.2.1. D=30

To highlight the superiority and robustness of the proposed EFPA-G algorithm, this section compares the convergence curves and search accuracy of the algorithm from two aspects. The computational accuracy results of the proposed EFPA-G algorithm and the compared algorithms are shown in Table 2. The mean and standard deviation are compared to reflect the robustness of the algorithms. Lower values of the mean and standard deviation indicate better robustness. The optimal results are highlighted in bold and marked. The convergence curves are shown in Figure 1 and Figure 2.

Based on the results from the 12 unimodal functions in Table 2, it is observed that the proposed EFPA-G algorithm can match 0.00E+00 (the optimal values) for all the six unimodal functions tested ($f_{1},f_{3},f_{6},f_{8},f_{9}$, and $f_{11}$).The other functions ($f_{2},f_{4},f_{7}$) are ahead of the other comparison algorithms even though they do not achieve the optimal values. However, in the $f_{10}$ function, the optimal values are obtained for the MPSO, MPA, and RACS algorithms, The mean reached −5.00E+00, and the standard deviations were 0.00E+00, 1.77E-12, and 6.26E-10, respectively. The presence of a large number of local extremes in the multimodal functions provides a better evaluation of the search performance of the proposed EFPA-G algorithm. From the experimental results of the 14 multimodal functions ($f_{13}$∼$f_{26}$) in Table 2, the performance of the proposed EFPA-G algorithm outperforms the other comparison algorithms obviously. For the five multimodal functions ($f_{14}$∼$f_{16}$, $f_{19}$, and $f_{22}$), the proposed EFPA-G algorithm is able to obtain the optimal values. The proposed EFPA-G algorithm obtains the first rank in both average and standard deviation for all the nine multimodal functions ($f_{13}$∼$f_{16}$, $f_{19}$, $f_{20}$, and $f_{22}$∼$f_{24}$). Therefore, in the testing of 26 benchmark functions, the mean and standard deviation of EFPA-G in 30 experiments were mostly optimal. This indicates that EFPA-G has superior robustness compared to the compared algorithms.

The convergence curves of the 10 unimodal functions in Figure 1 and Figure 2 show that the proposed EFPA-G algorithm outperforms the other compared algorithms in terms of convergence speed. The proposed EFPA-G algorithm converges much more quickly than the other compared algorithms in the seven unimodal functions ($f_{1}$∼$f_{4},f_{6},f_{8}$, and $f_{9}$). In the six multimodal functions ($f_{14}$∼$f_{16}$, $f_{19}$, $f_{23}$, and $f_{24}$), the convergence speed of the proposed EFPA-G algorithm obtains the first rank. In the other benchmark functions, the proposed EFPA-G algorithm obtained a satisfactory ranking.

From Table 2 and Figure 1 and Figure 2, it can be observed that the proposed EFPA-G algorithm exerts the exploitation capability in dealing with single-peaked functions and the exploration capability in dealing with multi-peaked functions.

#### 4.2.2. D=50

To test the sensitivity of the proposed EFPA-G algorithm in solving the same problem with different dimensions, this paper modifies the dimension *D* to 50 for the experiment, and the corresponding experimental results are shown in Table 3. Comparing the experimental results in Table 2 and Table 3.

For unimodal functions ($f_{1},f_{3},f_{6},f_{8},f_{9}$, and $f_{11}$) and multimodal functions ($f_{14}$∼$f_{16}$, $f_{19}$, and $f_{22}$) the optimal value of 0.00E+00 is achieved. The MPSO, MPA, and RACS algorithms all obtained optimal values for unimodal functions only ($f_{6},f_{10}$). Compared with the RACS algorithm, it outperforms the proposed EFPA-G algorithm in five benchmark functions ($f_{10}$, $f_{18}$∼$f_{20}$, and $f_{26}$). The MPA algorithm outperforms the proposed EFPA-G algorithm and other algorithms in functions ($f_{17},f_{21}$, and $f_{26}$) as the dimensionality increases. It is observed that with the increase in dimensions, the proposed EFPA-G algorithm can still obtain the first rank in most of the benchmark functions, including unimodal and multimodal functions.

Overall, the proposed algorithm EFPA-G still has stable performance and the best robustness in development and exploration with dimension D=50.

### 4.3. Simulation for WSN

In this section, two different WSN models of general-shape networks and O-shape networks are adopted to test the sensitivity of the proposed EFPA-G localization algorithm to different node distributions, as shown in Figure 3, both with 100 nodes in a 100 m × 100 m square area. Firstly, the localization accuracy and robustness of the proposed EFPA-G algorithm are demonstrated by a general simulation. Then, the localization accuracy of the proposed algorithm is tested for different anchor node proportions and node communication radii. The normalized root mean square error (NRMSE) and the mean distance error (MDE) are considered as metrics to evaluate the performance of the proposed algorithm, as shown in Equations (Equation 19) and (Equation 20).
(19)$$NRMSE=\frac{\sum_{i=1}^{N_{r}}\sqrt{{\left(x_{i}-{\hat{x}}_{i}\right)}^{2}+{\left(y_{i}-{\hat{y}}_{i}\right)}^{2}}}{N_{r}\times R}$$where ($x_{i}$, $y_{i}$) and (${\hat{x}}_{i}$, ${\hat{y}}_{i}$) denote the real and estimated coordinates of regular node *i*, respectively. *R* and $N_{r}$ denote the communication radius and the number of unknown nodes, respectively.
(20)$$MDE=\frac{\sum_{k=1}^{N_{a}}\sum_{i=1}^{N_{r}}\left|d_{ki}-{\hat{d}}_{ki}\right|}{N_{a}\times N_{r}}$$where $d_{ki}$ and ${\hat{d}}_{ki}$ are the real distance and estimated distance between anchor node *k* and unknown node *i*, respectively. $N_{a}$ denotes the number of anchor nodes.

**Figure 3 sensors-23-06463-f003:**
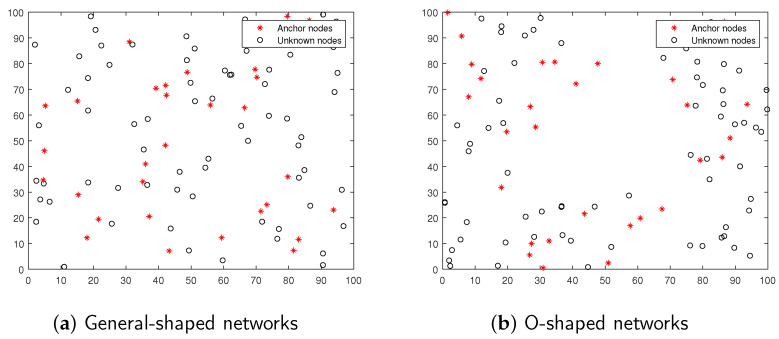
General-shaped and O-shaped networks.

#### 4.3.1. General Simulation

In this subsection, the proportion of anchor nodes is set to 30%, the number of unknown nodes is 70, and the node communication radius is 25 m. The experimental results are shown in Table 4, and the optimal solutions are marked in bold. The comparison results between the proposed EFPA-G and other comparison algorithms for the localization error of each unknown node are shown in Figure 4 and Figure 5.

From the experimental results in Table 4, it can be seen that under different node distribution conditions, the NRMSE and MDE obtained by the traditional DV-Hop positioning method achieved the worst positioning accuracy, of 31.2086% and 7.80E+00, respectively, which indicates that the performance of the metaheuristic optimization algorithm is better than the least squares method. For networks of different shapes, the EFPA-G algorithm obtains the first rank of mean and optimal values on NRMSE and MDE, with the lowest reaching 19.5182% and 4.88E+00. The “std” values of MPA and RACS are smaller than EFPA-G, which means that the former is more stable in 30 experiments. However, the proposed algorithm for node location pays more attention to positioning accuracy. Figure 4 and Figure 5 also intuitively observe that the EFPA-G algorithm has the smallest positioning error, especially at low anchor ratios.

#### 4.3.2. Effect of the Proportion of Anchor Nodes

In this subsection, the performance of the proposed algorithm is tested by gradually changing the proportion of anchor nodes (10%, 20%, 30%, and 40%) without changing the WSN model and the communication radius of the nodes. The experimental results are shown in Table 5. As the proportion of anchor nodes increases, the NRMSE of WSN node localization decreases, but this does not necessarily mean that a higher number of anchor nodes in the WSN model will result in higher localization accuracy. For example, in a general-shaped network, the NRMSE of the DV-Hop method increases from 30.12% to 30.48% as the proportion increases from 30% to 40%. In an O-shaped network, the NRMSE values of all metaheuristic localization methods first decrease and then increase as the proportion of anchor nodes increases. Overall, the proposed EFPA-G algorithm performs better in terms of overall performance for different networks when the proportion of anchor nodes is 30%.

#### 4.3.3. Effect of Node Communication Radius

The sensitivity of the proposed EFPA-G algorithm to different communication radii is tested by gradually changing the communication radius of the nodes (15 m, 20 m, 25 m, and 30 m) without changing the WSN model and the proportion of anchor nodes.

Table 6 shows the experimental results for different communication radii. As the communication radius of the nodes increases, the NRMSE values generally decrease in a general-shaped network. In an O-shaped network, the NRMSE values first decrease and then increase, indicating that blindly increasing the communication radius cannot guarantee an improvement in node localization accuracy. Instead, it places higher demands on hardware equipment and leads to wastage of resources. However, for both a general-shaped network and O-shaped network, the other compared methods obtain very close NRMSE values. This indicates that the FPA, CMFPA, MPA, and RACS algorithms have similar local exploitation capabilities in solving real optimization problems. The NRMSE obtained by the proposed EFPA-G algorithm ranked first with the changing radius of node communication, which suggests that the algorithm has superior performance in solving practical problems. Overall, EFPA-G outperforms state-of-the-art algorithms in terms of accuracy and robustness against network anisotropy.

## 5. Conclusions

This paper proposed a localization algorithm based on an enhanced flower pollination algorithm with Gaussian perturbation (EFPA-G) and the DV-Hop method. Firstly, the highly adaptive and parallel computing capabilities, and ease of implementation and application of the flower pollination algorithm were utilized to achieve sensor node localization. Subsequently, an enhanced strategy based on Lévy flight was proposed to address the problems of the difficulty in escaping from local extreme values, sensitivity to initial values, and possible over-exploration in the later iterations. By leveraging the diversity of the Lévy flight step length and the ability to avoid the influence of noisy data by jumping over long distances, the robustness of the flower pollination algorithm was improved. Then, a Gaussian perturbation strategy was proposed to address the critical imbalance between global exploration and local exploitation search capabilities of the flower pollination algorithm. The Gaussian distribution alters its curve according to the variation in the mean and variance, to improve the search capability, a large-scale population perturbation was implemented. Benchmark function experiments and WSN simulation experiments were conducted. The benchmark function experiments demonstrated that the proposed algorithm can achieve good search accuracy and convergence speed in different modes of benchmark functions. In the WSN simulation experiments, the proposed EFPA-G algorithm achieved lower localization error compared to similar algorithms such as PACS and MPA, which have been used in recent years, indicating better robustness. The main drawback of the proposed algorithm is that it depends on multiple complicated calculations, which may increase the overheads and power costs. Therefore, the proposed algorithm is more suitable for applications where there is little restriction of power consumption and overheads.

## Figures and Tables

**Figure 1 sensors-23-06463-f001:**
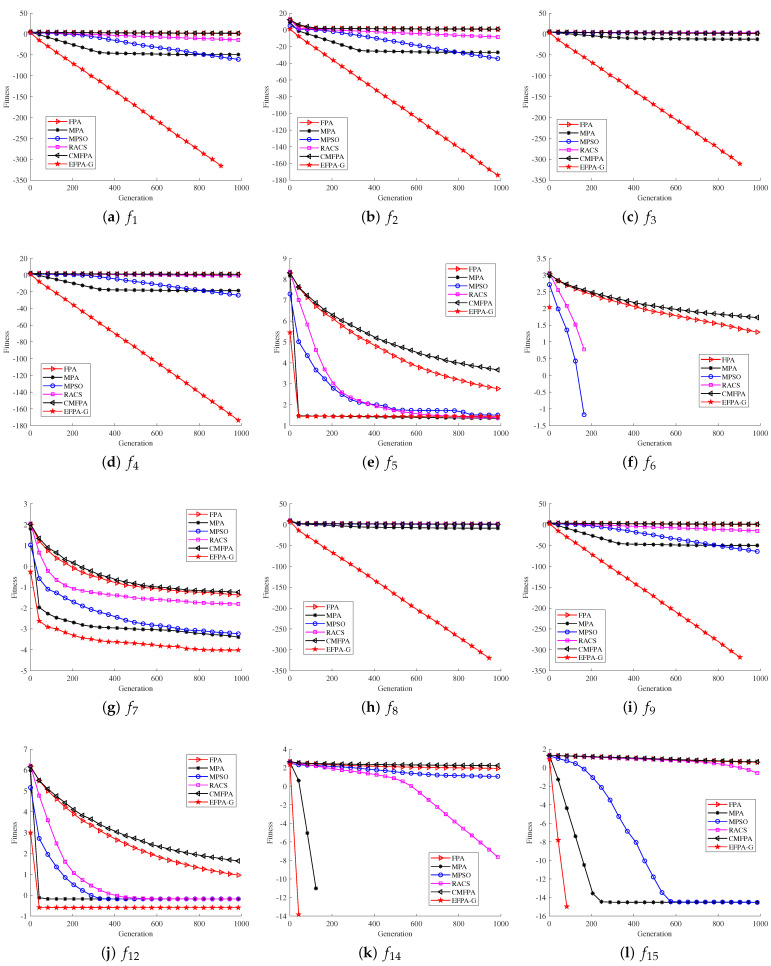
Comparison of proposed EFPA-G and other state-of-the-art algorithms for convergence performance (D=30).

**Figure 2 sensors-23-06463-f002:**
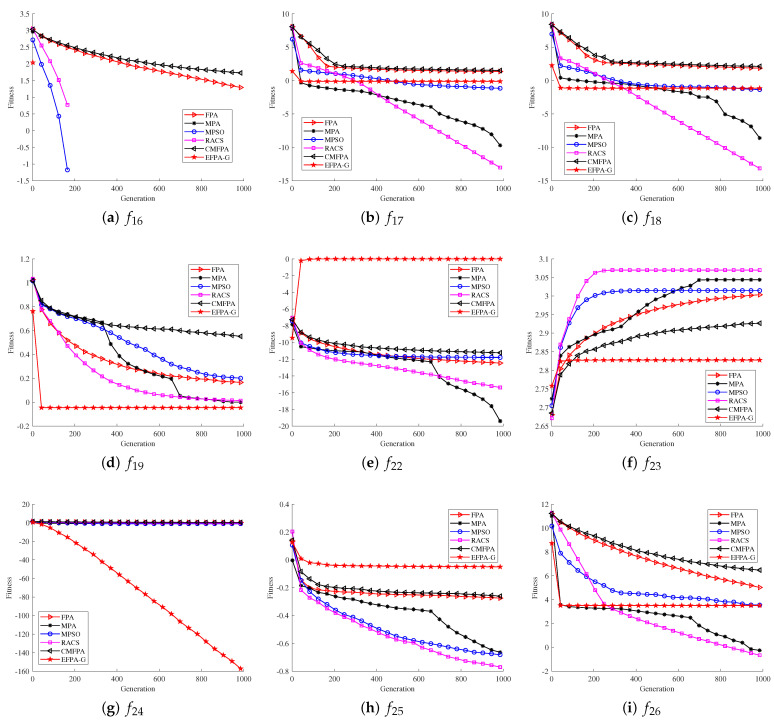
Comparison of proposed EFPA-G and other state-of-the-art algorithms for convergence performance (D=30).

**Figure 4 sensors-23-06463-f004:**
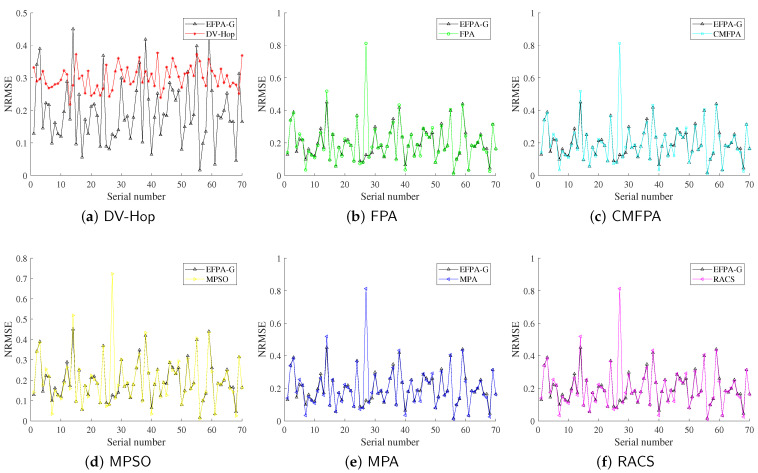
Comparison of proposed EFPA-G and other state-of-the-art algorithms for location error of general-shaped network.

**Figure 5 sensors-23-06463-f005:**
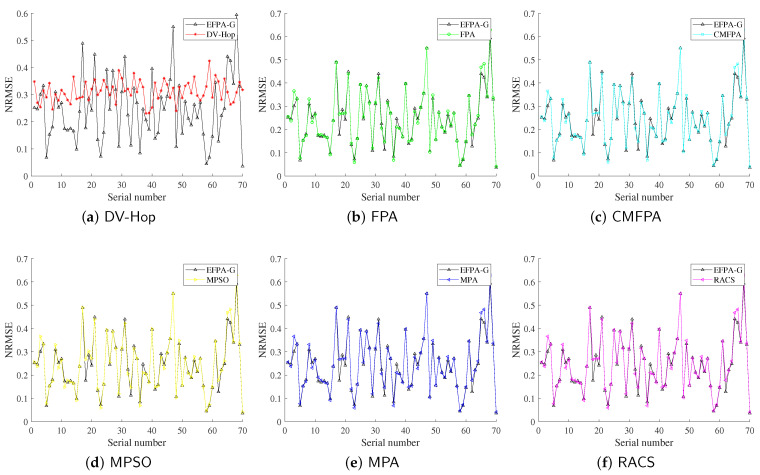
Comparison of proposed EFPA-G and other state-of-the-art algorithms for location error of O-shaped network.

**Table 1 sensors-23-06463-t001:** The information of the benchmark functions used in this paper.

Name	Function	Range	$f_{\mathrm{opt}}$
Sphere	$f_{1}=\sum_{i=1}^{D}x_{i}^{2}$	$\left[-100,100\right]$	0
Schwefel 2.22	$f_{2}=\sum_{i=1}^{D}\left|x_{i}\right|+\prod_{i=1}^{D}\left|x_{i}\right|$	$\left[-10,10\right]$	0
Schwefel 1.2	$f_{3}=\sum_{i=1}^{D}{\left(\sum_{j=1}^{i}x_{j}\right)}^{2}$	$\left[-100,100\right]$	0
Schwefel 2.21	$f_{4}=\max_{1\leq i\leq D}\left|x_{i}\right|$	$\left[-100,100\right]$	0
Rosenbrock	$f_{5}=\sum_{i=1}^{D-1}\left[100{\left(x_{i+1}-x_{i}^{2}\right)}^{2}+{\left(x_{i}-1\right)}^{2}\right]$	$\left[-30,30\right]$	0
Step	$f_{6}=\sum_{i=1}^{D}\left(\left\lfloor \left|x_{i}\right|\right\rfloor \right)$	$\left[-100,100\right]$	0
Quartic	$f_{7}=\sum_{i=1}^{D}ix_{i}^{4}+random\left[0,1\right)$	$\left[-1.28,1.28\right]$	0
Zakharov	$f_{8}=\sum_{i=1}^{D}x_{i}^{2}+{\left(0.5\sum_{i=1}^{D}ix_{i}\right)}^{2}+{\left(0.5\sum_{i=1}^{D}ix_{i}\right)}^{4}$	$\left[-5,10\right]$	0
Sum Squares	$f_{9}=\sum_{i=1}^{D}ix_{i}^{2}$	$\left[-10,10\right]$	0
Ridge	$f_{10}=x_{1}+d{\left(\sum_{i=2}^{D}x_{i}^{2}\right)}^{\alpha },d=1,\alpha =0.5$	$\left[-5,5\right]$	−5
Xin-She Yang 3	$f_{11}=e^{-\sum_{i=1}^{D}{\left(\frac{x_{i}}{\beta }\right)}^{2m}}-2e^{-\sum_{i=1}^{D}{\left(x_{i}\right)}^{2}}\cdot \prod_{i=1}^{D}\cos^{2}\left(x_{i}\right)$	$\left[-2\pi ,2\pi \right]$	−1
Dixon&Price	$f_{12}={\left(x_{1}-1\right)}^{2}+\sum_{i=2}^{D}i{\left(2x_{i}^{2}-x_{i-1}\right)}^{2}$	$\left[-10,10\right]$	0
Schwefel 2.26	$f_{13}=-\frac{1}{D}\sum_{i=1}^{D}x_{i}\sin\sqrt{\left|x_{i}\right|}$	$\left[-500,500\right]$	0
Rastrigin	$f_{14}=\sum_{i=1}^{D}\left[x_{i}^{2}-10\cos\left(2\pi x_{i}\right)+10\right]$	$\left[-5.12,5.12\right]$	0
Ackley 1	$f_{15}=-20e^{-0.02\sqrt{D^{-1}\sum_{i=1}^{D}x_{i}^{2}}}-e^{D^{-1}\sum_{i=1}^{D}\cos\left(2\pi x_{i}\right)}+20+e$	$\left[-35,35\right]$	0
Griewank	$f_{16}=\sum_{i=1}^{D}\frac{x_{i}^{2}}{4000}-\prod_{i=1}^{D}\cos\left(\frac{x_{i}}{\sqrt{i}}\right)+1$	$\left[-100,100\right]$	0
Generalized pen- alized function 1	$\begin{array}{c} f_{17}=\frac{\pi }{D}\left\{10\sin^{2}\pi x_{1}+\sum_{i=1}^{D}\left({\left(x_{i}-1\right)}^{2}\left(1+10\sin^{2}\pi x_{i}\right)\right)+{\left(x_{D}-1\right)}^{2}\right\}\\ +\sum_{i=1}^{D}u\left(x_{i},10,100,4\right) \end{array}$	$\left[-50,50\right]$	0
Generalized pen- alized function 2	$\begin{array}{c} f_{18}=\frac{1}{10}\left\{\sin^{2}3\pi x_{1}+\sum_{i=1}^{D-1}\left({\left(x_{i}-1\right)}^{2}\left(1+\sin^{2}3\pi x_{i+1}\right)\right)\right\}+\\ \frac{1}{10}\left\{\left(x_{D}-1\right){\left(1+\sin2\pi x_{D}\right)}^{2}\right\}+\sum_{i=1}^{D}u\left(x_{i},5,100,4\right) \end{array}$	$\left[-50,50\right]$	0
Periodic	$f_{19}=1+\sum_{i=1}^{D}\sin^{2}\left(x_{i}\right)-0.1e^{-\sum_{i=1}^{D}x_{i}^{2}}$	$\left[-10,10\right]$	0.9
Ackley 4	$f_{20}=\sum_{i=1}^{D-1}\left(e^{-0.2}\sqrt{x_{i}^{2}+x_{i+1}^{2}}+3\left(\cos\left(2x_{i}\right)+\sin\left(2x_{i+1}\right)\right)\right)$	$\left[-10,10\right]$	−4.59
Xin-She Yang 2	$f_{21}=\left(\sum_{i=1}^{D}\left|x_{i}\right|\right)e^{-\sum_{i=1}^{D}\sin\left(x_{i}^{2}\right)}$	$\left[-2\pi ,2\pi \right]$	0
Xin-She Yang 4	$f_{22}=\left[\sum_{i=1}^{D}\sin^{2}\left(x_{i}\right)-e^{-\sum_{i=1}^{D}x_{i}^{2}}\right]e^{-\sum_{i=1}^{D}\sin^{2}\sqrt{\left|x_{i}\right|}}$	$\left[-10,10\right]$	−1
Styblinski-Tank	$f_{23}=\frac{1}{2}\sum_{i=1}^{D}\left(x_{i}^{4}-16x_{i}^{2}+5x_{i}\right)$	$\left[-5,5\right]$	0
Salomon	$f_{24}=1-\cos\left(2\pi \sqrt{\sum_{i=1}^{D}x_{i}^{2}}\right)+0.1\sqrt{\sum_{i=1}^{D}x_{i}^{2}}$	$\left[-100,100\right]$	−39.16599·D
Happy Cat	$f_{25}={\left[{\left({∥x∥}^{2}-D\right)}^{2}\right]}^{\alpha }+\frac{1}{D}\left(\frac{1}{2}{∥x∥}^{2}+\sum_{i=1}^{D}x_{i}\right)+\frac{1}{2},\alpha =0.5$	$\left[-2,2\right]$	0
Qing	$f_{26}=\sum_{i=1}^{D}{\left(x_{i}^{2}-i\right)}^{2}$	$\left[-500,500\right]$	0

**Table 2 sensors-23-06463-t002:** Results for unimodal and multimodal functions (D=30).

Function		Algorithm						Function		Algorithm					
		FPA	CMFPA	MPSO	MPA	RACS	EFPA-G			FPA	CMFPA	MPSO	MPA	RACS	EFPA-G
$f_{1}$	Ave	2.11E+01	1.19E+02	6.69E-62	9.61E-50	8.34E-15	**0.00E+00**	$f_{14}$	Ave	8.45E+01	1.68E+02	1.17E+01	0.00E+00	1.19E-08	**0.00E+00**
	Std	1.26E+01	7.99E+01	3.66E-61	1.05E-49	6.55E-15	**0.00E+00**		Std	1.38E+01	1.39E+01	1.32E+01	0.00E+00	8.19E-09	**0.00E+00**
$f_{2}$	Ave	6.21E+00	1.00E+01	1.39E-35	1.33E-27	2.75E-09	**1.77E-178**	$f_{15}$	Ave	3.81E+00	4.26E+00	2.96E-15	3.08E-15	1.78E-01	**0.00E+00**
	Std	2.03E+00	2.06E+00	4.54E-35	1.72E-27	8.79E-10	**0.00E+00**		Std	1.03E+00	8.13E-01	1.35E-15	1.23E-15	4.26E-01	**0.00E+00**
$f_{3}$	Ave	1.86E+01	8.07E+01	2.60E+02	7.83E-13	1.16E+03	**0.00E+00**	$f_{16}$	Ave	6.30E-01	1.02E+00	0.00E+00	0.00E+00	1.53E-13	**0.00E+00**
	Std	9.71E+00	3.31E+01	1.54E+02	2.82E-12	5.61E+02	**0.00E+00**		Std	1.40E-01	4.63E-02	0.00E+00	0.00E+00	6.53E-13	**0.00E+00**
$f_{4}$	Ave	7.71E+00	9.89E+00	3.13E-25	2.70E-19	1.30E-01	**7.58E-177**	$f_{17}$	Ave	2.26E+01	3.14E+01	6.54E-02	5.95E-11	**5.02E-14**	7.64E-01
	Std	1.49E+00	1.75E+00	7.68E-25	2.34E-19	3.43E-02	**0.00E+00**		Std	7.89E+00	1.54E+01	1.24E-01	2.55E-11	**3.08E-14**	4.24E-01
$f_{5}$	Ave	5.45E+02	4.47E+03	3.19E+01	2.24E+01	2.38E+01	2.78E+01	$f_{18}$	Ave	6.96E+01	1.29E+02	4.33E-02	7.40E-10	**3.34E-14**	7.22E-02
	Std	4.30E+02	3.15E+03	3.17E+01	6.19E-01	3.96E+00	4.98E+00		Std	3.36E+01	8.84E+01	3.80E-02	3.21E-10	**3.06E-14**	3.70E-02
$f_{6}$	Ave	1.87E+01	5.26E+01	**0.00E+00**	**0.00E+00**	**0.00E+00**	**0.00E+00**	$f_{19}$	Ave	1.46E+00	3.51E+00	1.59E+00	1.00E+00	1.03E+00	**9.00E-01**
	Std	5.46E+00	1.48E+01	**0.00E+00**	**0.00E+00**	**0.00E+00**	**0.00E+00**		Std	6.81E-02	5.06E-01	5.13E-01	2.01E-03	6.42E-03	**4.52E-16**
$f_{7}$	Ave	4.27E-02	5.71E-02	5.73E-04	4.08E-04	1.53E-02	**9.56E-05**	$f_{20}$	Ave	1.15E+01	2.14E+01	−3.93E+01	−4.71E+01	−4.81E+01	**−4.69E+01**
	Std	1.36E-02	1.90E-02	2.95E-04	2.30E-04	4.10E-03	**6.88E-05**		Std	9.23E+00	9.55E+00	3.02E+00	1.94E+00	1.21E+00	**4.47E+00**
$f_{8}$	Ave	4.70E+00	1.03E+01	6.02E+00	4.05E-09	3.81E+01	**0.00E+00**	$f_{21}$	Ave	5.87E-12	6.34E-12	4.23E-11	7.30E-12	**3.66E-12**	7.98E-11
	Std	2.84E+00	5.64E+00	8.05E+00	4.15E-09	1.27E+01	**0.00E+00**		Std	2.32E-12	2.91E-12	4.29E-11	2.84E-12	**2.07E-13**	2.30E-10
$f_{9}$	Ave	2.20E+00	1.37E+01	4.78E-66	1.68E-50	7.77E-16	**0.00E+00**	$f_{22}$	Ave	3.35E-13	5.58E-12	1.62E-12	7.93E-21	3.59E-16	**−1.00E+00**
	Std	1.08E+00	6.90E+00	1.09E-65	3.13E-50	3.31E-16	**0.00E+00**		Std	5.79E-14	1.57E-12	1.43E-12	3.42E-21	1.72E-16	**0.00E+00**
$f_{10}$	Ave	−4.59E+00	−3.85E+00	**−5.00E+00**	**−5.00E+00**	**−5.00E+00**	−1.50E+00	$f_{23}$	Ave	−1.01E+03	−8.47E+02	−1.03E+03	−1.11E+03	−1.17E+03	**−6.73E+02**
	Std	1.09E-01	2.63E-01	**0.00E+00**	**1.77E-12**	**6.26E-10**	8.97E-01		Std	1.27E+01	2.47E+01	3.96E+01	2.12E+01	1.79E-13	**3.93E+01**
$f_{11}$	Ave	−2.56E-01	−2.98E-01	9.95E-01	1.99E-01	9.95E-01	**−1.00E+00**	$f_{24}$	Ave	2.49E+00	2.55E+00	9.99E-02	1.10E-01	3.55E-01	**1.33E-159**
	Std	8.96E-01	8.64E-01	3.39E-16	9.91E-01	3.39E-16	**0.00E+00**		Std	3.29E-01	4.23E-01	6.30E-09	3.05E-02	4.97E-02	**7.29E-159**
$f_{12}$	Ave	8.63E+00	4.28E+01	6.67E-01	6.67E-01	6.67E-01	2.55E-01	$f_{25}$	Ave	5.31E-01	5.48E-01	2.06E-01	2.16E-01	**1.68E-01**	8.92E-01
	Std	3.97E+00	2.59E+01	1.96E-07	1.06E-09	4.35E-05	6.84E-03		Std	9.33E-02	6.84E-02	7.30E-02	4.65E-02	**2.33E-02**	2.02E-01
$f_{13}$	Ave	−2.70E+02	−1.79E+02	−3.15E+02	−3.36E+02	−4.19E+02	**−1.41E+02**	$f_{26}$	Ave	9.19E+04	2.88E+06	3.52E+03	5.40E-01	**1.98E-01**	3.24E+03
	Std	6.91E+00	7.94E+00	2.01E+01	1.71E+01	1.00E+00	**2.09E+01**		Std	7.72E+04	3.60E+06	1.11E+04	1.66E+00	**8.29E-02**	8.43E+02

Bold values indicate that the function is optimal in experimental comparisons.

**Table 3 sensors-23-06463-t003:** Results for unimodal and multimodal functions (D=50).

Function		Algorithm						Function		Algorithm					
		FPA	CMFPA	MPSO	MPA	RACS	EFPA-G			FPA	CMFPA	MPSO	MPA	RACS	EFPA-G
$f_{1}$	Ave	3.50E+02	1.26E+03	1.43E-45	3.84E-46	7.23E-08	**0.00E+00**	$f_{14}$	Ave	1.78E+02	3.33E+02	1.53E+01	0.00E+00	1.40E+00	**0.00E+00**
	Std	1.25E+02	4.98E+02	7.41E-45	7.34E-46	2.93E-08	**0.00E+00**		Std	1.97E+01	2.21E+01	3.36E+01	0.00E+00	9.42E-01	**0.00E+00**
$f_{2}$	Ave	1.61E+01	2.20E+01	5.64E-32	3.88E-26	5.07E-05	**1.80E-178**	$f_{15}$	Ave	4.49E+00	4.35E+00	4.19E-01	3.32E-15	1.23E+01	**0.00E+00**
	Std	1.70E+00	2.81E+00	2.11E-31	4.70E-26	1.03E-05	**0.00E+00**		Std	1.22E+00	1.06E+00	1.91E+03	9.01E-16	4.33E+00	**0.00E+00**
$f_{3}$	Ave	3.77E+02	7.17E+02	2.84E+03	2.48E-08	1.17E+04	**0.00E+00**	$f_{16}$	Ave	1.08E+00	1.35E+00	0.00E+00	0.00E+00	1.26E-08	**0.00E+00**
	Std	1.22E+02	2.57E+02	1.91E+03	1.02E-07	1.94E+03	**0.00E+00**		Std	2.88E-02	9.60E-02	0.00E+00	0.00E+00	1.83E-08	**0.00E+00**
$f_{4}$	Ave	1.25E+01	1.35E+01	3.54E-22	1.36E-17	2.33E+00	**1.74E-175**	$f_{17}$	Ave	3.23E+01	3.58E+01	2.50E+00	**2.52E-09**	9.05E-07	1.63E+00
	Std	1.91E+00	1.58E+00	1.39E-21	9.06E-18	4.29E-01	**0.00E+00**		Std	8.86E+00	1.56E+01	3.24E+00	**6.82E-10**	5.27E-07	6.35E-01
$f_{5}$	Ave	1.26E+04	1.03E+05	4.92E+03	4.34E+01	5.76E+01	4.85E+01	$f_{18}$	Ave	1.51E+02	1.85E+02	1.22E+00	4.90E-03	**3.59E-07**	1.12E-01
	Std	8.62E+03	6.20E+04	2.00E+04	5.44E-01	2.00E+01	6.88E-03		Std	6.45E+01	7.93E+01	1.41E+00	7.86E-03	**1.22E-07**	5.15E-02
$f_{6}$	Ave	9.58E+01	1.71E+02	**0.00E+00**	**0.00E+00**	**0.00E+00**	**0.00E+00**	$f_{19}$	Ave	3.11E+00	6.80E+00	2.30E+00	1.01E+00	1.35E+00	**9.00E-01**
	Std	1.38E+01	2.75E+01	**0.00E+00**	**0.00E+00**	**0.00E+00**	**0.00E+00**		Std	2.02E-01	6.73E-01	6.13E-01	7.12E-03	5.44E-02	**4.52E-16**
$f_{7}$	Ave	1.52E-01	1.81E-01	8.88E-04	6.65E-04	3.05E-02	**8.43E-05**	$f_{20}$	Ave	5.09E+01	7.40E+01	5.64E-32	−7.44E+01	**−6.85E+01**	−7.95E+01
	Std	5.13E-02	5.37E-02	3.87E-04	3.84E-04	6.91E-03	**9.02E-05**		Std	1.37E+01	1.17E+01	6.51E+00	2.74E+00	**1.45E+00**	6.96E+00
$f_{8}$	Ave	5.52E+01	8.81E+01	9.81E+01	6.66E-05	2.88E+02	**0.00E+00**	$f_{21}$	Ave	**1.96E-20**	**1.92E-20**	8.90E-17	**1.97E-20**	**3.71E-20**	9.85E-19
	Std	1.66E+01	2.01E+01	5.53E+01	6.29E-05	5.32E+01	**0.00E+00**		Std	**8.23E-21**	**7.09E-21**	4.87E-16	**6.74E-21**	**2.56E-21**	2.37E-18
$f_{9}$	Ave	6.94E+01	2.37E+02	9.73E-56	1.77E-46	1.43E-08	**0.00E+00**	$f_{22}$	Ave	5.31E-21	3.44E-19	3.35E-20	3.93E-22	9.24E-22	**−1.00E+00**
	Std	2.26E+01	7.41E+01	5.32E-55	2.91E-46	4.21E-09	**0.00E+00**		Std	5.86E-22	1.32E-19	5.13E-20	2.15E-21	1.15E-22	**0.00E+00**
$f_{10}$	Ave	−3.47E+00	−1.48E+00	**−5.00E+00**	**−5.00E+00**	**−5.00E+00**	−1.41E+00	$f_{23}$	Ave	−1.57E+03	−1.25E+03	−1.71E+03	−1.77E+03	−1.96E+03	−1.07E+03
	Std	2.93E-01	5.64E-01	**1.65E-16**	**1.77E-11**	**2.99E-06**	1.01E+00		Std	2.42E+01	4.17E+01	4.41E+01	3.19E+01	2.41E-07	7.78E+01
$f_{11}$	Ave	−1.83E-01	6.93E-02	9.92E-01	−5.22E-02	9.92E-01	**−1.00E+00**	$f_{24}$	Ave	4.66E+00	4.84E+00	9.99E-02	1.43E-01	7.64E-01	**3.57E-159**
	Std	8.98E-01	8.79E-01	3.39E-16	9.94E-01	3.39E-16	**0.00E+00**		Std	5.32E-01	5.70E-01	1.26E-08	5.04E-02	7.15E-02	**1.90E-158**
$f_{12}$	Ave	2.26E+02	1.14E+03	5.24E+00	**6.67E-01**	**6.80E-01**	**2.56E-01**	$f_{25}$	Ave	6.41E-01	6.84E-01	4.38E-01	4.46E-01	3.75E-01	9.90E-01
	Std	1.56E+02	4.97E+02	2.51E+01	**3.00E-09**	**2.79E-02**	**1.49E-02**		Std	6.72E-02	5.15E-02	8.45E-02	5.99E-02	4.25E-02	2.73E-01
$f_{13}$	Ave	−2.39E+02	−1.41E+02	−2.64E+02	−3.22E+02	−4.18E+02	−1.55E+02	$f_{26}$	Ave	9.06E+06	8.09E+07	2.31E+06	**8.69E+01**	**9.28E+01**	1.87E+04
	Std	7.17E+00	7.54E+00	2.58E+01	1.27E+01	1.27E+00	5.49E+01		Std	5.34E+06	5.29E+07	8.35E+06	**1.25E+02**	**2.08E+01**	2.11E+03

Bold values indicate that the function is optimal in experimental comparison results.

**Table 4 sensors-23-06463-t004:** Comparison of proposed EFPA-G and other state-of-the-art algorithms for location error.

	General-Shaped Network				O-Shaped Network			
	Algorithm	Average	Best	Std	Algorithm	Average	Best	Std
Normalized Root Mean Square Error (NRMSE %)	DV-Hop	30.1224%	24.4230%	2.94E-02	DV-Hop	31.2086%	25.3410%	3.23E-02
	FPA	20.2731%	20.2595%	1.86E-04	FPA	24.9798%	24.9442%	3.60E-04
	CMFPA	20.2648%	20.2629%	8.87E-06	CMFPA	25.0067%	24.9509%	3.25E-04
	MPSO	20.1720%	19.3034%	3.39E-03	MPSO	25.0103%	24.8910%	4.42E-04
	MPA	20.2650%	20.2649%	6.17E-08	MPA	25.0280%	25.0280%	**4.20E-07**
	RACS	20.2650%	20.2650%	**2.71E-15**	RACS	25.0159%	24.9552%	2.76E-04
	EFPA-G	**19.5182%**	**19.0583%**	2.21E-03	EFPA-G	**24.5535%**	**24.0744%**	1.56E-03
Mean Distance Error (MDE)	DV-Hop	7.53E+00	6.11E+00	7.35E-01	DV-Hop	7.80E+00	6.34E+00	8.07E-01
	FPA	5.07E+00	5.06E+00	4.64E-03	FPA	6.24E+00	6.24E+00	9.01E-03
	CMFPA	5.07E+00	5.07E+00	2.22E-04	CMFPA	6.25E+00	6.24E+00	8.13E-03
	MPSO	5.04E+00	4.83E+00	8.48E-02	MPSO	6.25E+00	6.22E+00	1.10E-02
	MPA	5.07E+00	5.07E+00	1.54E-06	MPA	6.26E+00	6.26E+00	**1.05E-05**
	RACS	5.07E+00	5.07E+00	**6.79E-14**	RACS	6.25E+00	6.24E+00	6.91E-03
	EFPA-G	**4.88E+00**	**4.76E+00**	5.52E-02	EFPA-G	**6.14E+00**	**6.02E+00**	3.90E-02

Bold values indicate that the algorithm achieves the best in the experimental comparison results.

**Table 5 sensors-23-06463-t005:** Results of different proportions of anchor nodes.

	Algorithm	Proportion of Anchor Nodes (%)			
		10	20	30	40
General-shaped network	DV-Hop	39.38%	31.29%	30.12%	30.48%
	FPA	26.85%	20.50%	20.27%	18.97%
	CMFPA	26.84%	**20.44%**	20.27%	18.97%
	MPSO	26.85%	20.49%	20.17%	18.47%
	MPA	26.84%	**20.44%**	20.27%	18.97%
	RACS	26.84%	**20.44%**	20.27%	18.97%
	EFPA-G	**25.94%**	20.45%	**19.52%**	**18.02%**
O-shaped network	DV-Hop	37.66%	31.49%	31.21%	29.70%
	FPA	34.14%	25.50%	24.98%	25.79%
	CMFPA	33.91%	25.51%	25.01%	25.79%
	MPSO	34.10%	25.52%	25.01%	25.78%
	MPA	33.92%	25.51%	25.03%	25.79%
	RACS	33.97%	25.51%	25.02%	25.79%
	EFPA-G	**33.41%**	**25.45%**	**24.56%**	**25.14%**

Bold values indicate that the algorithm achieves the best in the experimental comparison results.

**Table 6 sensors-23-06463-t006:** Results of different node communication radius.

	Algorithm	Communication Radius of Nodes (m)			
		15	20	25	30
General-shaped network	DV-Hop	158.78%	43.64%	30.12%	29.42%
	FPA	72.53%	22.31%	20.27%	21.74%
	CMFPA	71.83%	22.31%	20.27%	21.74%
	MPSO	71.90%	22.68%	20.17%	21.58%
	MPA	71.83%	22.31%	20.27%	21.74%
	RACS	71.84%	22.31%	20.26%	21.74%
	EFPA-G	**71.43%**	**22.14%**	**19.52%**	**20.98%**
O-shaped network	DV-Hop	156.64%	42.18%	31.21%	29.71%
	FPA	48.12%	38.45%	24.98%	23.16%
	CMFPA	48.12%	38.22%	25.01%	23.16%
	MPSO	47.97%	39.05%	25.01%	23.17%
	MPA	48.12%	38.23%	25.03%	23.15%
	RACS	48.12%	38.12%	25.02%	23.16%
	EFPA-G	**46.53%**	**37.17%**	**24.55%**	**22.33%**

Bold values indicate that the algorithm achieves the best in the experimental comparison results.

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
