# Peer review of "An Enhanced Flower Pollination Algorithm with Gaussian Perturbation for Node Location of a WSN"

_sensors, 2023, doi:10.3390/s23146463_

Round 1
Reviewer 1 Report
1. The authors must justify the significance of the DV-Hop method discussed in Section 1: Background.
2. The authors should clarify the objective function defined on page 6 in terms of
a. Is (x, y) the estimated or actual coordinate of the unknown node?
b. Line no. 203-204 states,” localization problem is transformed into the problem of minimizing the difference between estimated and actual locations” but Eq. (18) computes the difference of the distance between the anchor node and unknown node.
3. Will step 7 of Algorithm 1 be ever executed (as P = 0.5 in section 3.1 Parameter settings on page 6)?
4. Step 12 of Algorithm 1 is incomplete.
5. Include a proper justification for the value of “Std.” of MPA and RACS for the General-shaped network in Table 4.
6. The authors should illustrate the robustness of the proposed approach precisely.
7. Is the EFPA-G algorithm executed by the sensor node? Justify.
8. The complexity of the approach may be discussed.
9. The variables used in Eq. 19 and 20 should be defined.
10. Are Normalized Error in Figure 3 and NRMSE in Eq. 19 the same?
11. Section 3.3 needs to be improved with concise and precise discussion.
12. MPSO in line no. 317 on page 12 shall be replaced by MPA.
The uuality of English is satisfactory.
Reviewer 2 Report
Title: A Enhanced Flower Pollination Algorithm with Gaussian Perturbation for Nodes Location of WSN
Comments:
Please avoid very long sentences such as “Localization, one of the essential problems in Internet of Things (IoT) and Wireless Sensor Network (WSN) applications, remains to be challenged in numerous ways. Most traditional range-free localization algorithms cannot fulfill the practical demand for high localization accuracy”
Justify using enhanced flower pollination algorithm (FPA)
Added some numerical results in abstract.
Section 3 Experimental results and discussion. The word “Experimental” related to hardware. So please change it into numerical.
What do you mean by Numerical experiment?
Please highlight the main findings from studying the effect of the proportion of anchor nodes
Please highlight the main findings from studying the effect of node communication radius
Conclusion is similar to abstract. Please revise it and highlight the main findings
Moderate editing of English language required
Reviewer 3 Report
The uthors motivation is clear, easy to read. Th article follows a logical way.
The authors used 33 relevant literature. The literatures are coming after 2010. Good job!
The figures quality is perfect but fiugre 1. is too much. After zoom on screen the quality is perfect, but if somebody use a printed version during his/her work impossible to see it. Maybe it is neccessary to modify or redesign this part.
1.2. Flower pollination algorithm - need more explenation
Good job! Nothing to say!
Round 2
Reviewer 2 Report
The paper can be accepted
Moderate editing of English language required